# The Social Distance Impacts from COVID-19 Pandemic on the Development of Two Orders of a Concurrent Training Programme for Morbidly Obese Patients

**DOI:** 10.3390/ijerph192013408

**Published:** 2022-10-17

**Authors:** Pedro Delgado-Floody, Luis Chirosa-Ríos, Iris Paola Guzmán-Guzmán, Claudia Andrea Vargas, Karina Sandoval-Aguilera, Felipe Caamaño-Navarrete, Francisco Guede-Rojas, Cristian Alvarez

**Affiliations:** 1Department of Physical Education, Sport and Recreation, Universidad de La Frontera, Temuco 4811230, Chile; 2Department Physical Education and Sports, Faculty of Sport Sciences, University of Granada, 18011 Granada, Spain; 3Strength & Conditioning Laboratory, CTS-642 Research Group, Department Physical Education and Sports, Faculty of Sport Sciences, University of Granada, 18011 Granada, Spain; 4Faculty of Chemical-Biological Sciences, Universidad Autónoma de Guerrero, Chilpancingo 39087, Mexico; 5Physical Education Career, Universidad Autónoma de Chile, Temuco 4780000, Chile; 6Exercise and Rehabilitation Sciences Institute, School of Physical Therapy, Faculty of Rehabilitation Sciences, Universidad Andres Bello, Santiago de Chile 7591538, Chile

**Keywords:** concurrent training, morbid obesity, metabolic syndrome, cardiometabolic risk factor, high-intensity interval training, resistance training

## Abstract

Background: Although there is relevant information regarding the consequences of the coronavirus SARS-CoV-2 (COVID-19), little is known about the impact of the imposed social confinement (at home) on the development of exercise training programmes in populations with morbid obesity. Aim: To describe the effects of the imposed COVID-19 confinement on the cardiometabolic health benefits acquired through a concurrent training programme that started before the pandemic in populations with morbid obesity. Methods: This was an experimental randomized clinical study, in which sedentary morbidly obese women were assigned 1:1 to a high-intensity interval training (HIIT) plus resistance training (RT) group (HIIT + RT; *n* = 11; BMI 42.1 ± 6.6) or to the same exercise dose, but in different order group of RT plus HIIT group (RT + HIIT; *n* = 7; BMI 47.5 ± 8.4). Both groups undertook two sessions/week. When COVID-19 confinement at home started, a post-test was applied in January 2020 (Post1) and after 20 months (Post2). The main outcomes were waist circumference (WC), systolic (SBP) and diastolic blood pressure (DBP), high-density lipids (HDL-c), triglycerides (Tg), and fasting plasma glucose (FPG). Results: In the HIIT + RT group, the WC showed significant increases from Post1 to Post2 (Δ + 3.1 cm, *p* = 0.035); in the RT + HIIT group, it decreased from Post1 to Post2 (Δ − 4.8 cm, *p* = 0.028). In the HIIT + RT group, SBP showed significant increases from Post1 to Post2 (Δ + 6.2 mmHg, *p* = 0.041); the RT + HIIT group decreased SBP from Pre0 to Post1 (Δ − 7.2 mmHg, *p* = 0.026) and increased DBP from Pre0 to Post1 (Δ + 8.1 mmHg, *p* = 0.015). Tg in the HIIT + RT group decreased from Pre0 to Post1 (Δ − 40.1 mg/dL, *p* = 0.023) but increased from Post1 to Post2 (Δ + 86.3 mg/dL, *p* < 0.0001). Conclusions: The COVID-19 social confinement worsened metabolic syndrome (MetS) outcomes that had improved from 20 weeks’ RT + HIIT during the training period, such as WC, SBP, and Tg from HIIT + RT, when, worryingly, SBP increased to another more serious clinical classification in both groups.

## 1. Introduction

Physical inactivity (defined as not adhering to international physical activity (PA) guidelines [1]) is the fourth-largest cause of mortality in the world [2]. Likewise, the global burden related to physical inactivity is significant (7.2% of all causes of death are attributed to the effects of physical inactivity) [3]. Other habits, such as sedentary behaviour [4] and unhealthy nutrition [5], are major modifiable lifestyle behaviours that promote obesity, a disease that was known in the past as the first global pandemic [6]. However, from the start of the pandemic due to the severe acute respiratory syndrome coronavirus 2 (SARS-CoV-2) causing of the called coronavirus disease 2019 (COVID-19), humans have been suffering multiple consequences at the level of their ‘physical’ and ‘mental’ health [7]; the social distance actions such as ‘at-home confinement’ have, on one hand, favoured the virus’s control [8] but, unfortunately, on the other hand, have exacerbated sedentary behaviour [9] and thus the prevalence of physical inactivity among children, adolescents, adults, and older persons [10,11,12]. Moreover, more risk factors for metabolic syndrome (MetS) have been reported during at-home confinement, during which members of the population have additionally increased their energy consumption [13]. These habits have been promoted, for example, by several increases in food delivery services through mobile APPs during the COVID-19 pandemic [14,15] but combined with low possibilities for energy expenditure at home. The above confinement situations have caused a worsening of the population’s overall cardiometabolic health and increased the risk of suffering MetS in at-risk populations. For example, it has been reported that members of the adult population increased their screen time (6.79 versus 5.06 h/day) from the pre-pandemic situation, and they have been strongly recommended to recover and increase their PA patterns [16]. Other detrimental reports during the COVID-19 pandemic include an increase in body mass (79.7 kg to 81.4 kg) [17] and the body mass index (BMI) (+0.15 (kg/m^2^)) due to COVID-19 [18]. Similarly, another study showed an incremented fasting plasma glucose (+8.5 mg/dL), being also glucose control affected during confinement at home [17]. Furthermore, Laffin et al., reported a rise in blood pressure in the adult population during the COVID-19 pandemic [19]. Another study conducted with adults showed that their 24 h systolic/diastolic blood pressure was significantly higher during the COVID-19 pandemic than in the pre-pandemic stage [20]. Other studies have revealed sharp high-density lipoprotein cholesterol (HDL-c) decreases [21] as well as increases in triglycerides [22], thus exacerbating the prevalence of MetS [23]. In addition, around the world, it has been reported a sharp decrease in the PA patterns globally. However, worryingly early evidence from the start of the pandemic reported that these reductions in PA were more pronounced in Latin America compared with European countries [10,24,25]. Thus, considering that Latin America is a region characterized by several countries with wide social inequalities, poor healthcare access, and a high prevalence of cardiometabolic diseases, such as type 2 diabetes mellitus and arterial hypertension [26], reporting information from those populations that are more affected by COVID-19 and face higher MetS risk could increase the possibilities of implementing preventive strategies in a future post-pandemic state/scenarios. In this sense, Polero et al., reported that the population reduced their PA levels which evidently increased their cardiometabolic risk or disease [27]. Likewise, a systematic literature review indicated that PA levels decreased as a result of COVID-19 confinement [28].

A lifestyle change program is characterized by the inclusion of regular exercise, and additionally other specific behaviours such as nutrition, and sleep patterns, among others (tobacco, and alcohol consumption) [29,30]. Regular exercise plays a critical role for some populations, such as morbidly obese candidates for bariatric surgery [31]. Thus, in addition to improving cardiometabolic risk factors for MetS, exercise training, such as concurrent training (CT, a combination of the same exercise session of resistance plus other endurance/aerobic or high-intensity interval training methodologies), promotes several other benefits, including better and healthier conditions for bariatric surgery and preparation for post-surgery challenges in the recovery period, with major success [32,33]. Unfortunately, little is known regarding the impact of the COVID-19 confinement on the cardiometabolic health benefits acquired before the pandemic through exercise training programmes such as CT in populations with morbid obesity. Along this line, it is relevant to mention that several countries and their health/sports systems were developing different exercise training programmes and training regimes at different levels (i.e., recreational, competitive, and professional) when the COVID-19 pandemic and confinement actions abruptly started. This situation meant that several exercise training programmes ceased and thus a high number of participants lost the exercise benefits that they had acquired before confinement. Hence, we hypothesized that 20 months of social distancing due to the COVID-19 pandemic affected the benefit of a concurrent training programme in morbidly obese patients. We aimed to describe the effects of the imposed COVID-19 social confinement on the cardiometabolic health benefits acquired through a CT programme started before the pandemic in populations with morbid obesity.

## 2. Materials and Methods

### 2.1. Participants

This study was a randomized controlled trial in which (*n* = 34) women with morbid obesity from the Morbid Obesity Association of Temuco City, Chile, were invited to participate by a public call and contacted by phone directly to all those interested. The participants were invited to participate in the interventions since they were part of a previous recommended lifestyle exercise programme in the same institution [32]. All the participants were informed of the pre–post procedures and of the possible risks/benefits potentially involved in participation in the study, after which they signed an informed consent form. The study was undertaken in accordance with the Declaration of Helsinki (2013) and has been approved by the Ethical Committee of the Universidad de La Frontera, Temuco, Chile (ACTA No 080_21).

The sample size was calculated using the G*Power version 3.1.9.7, and the observed delta changes in fasting plasma glucose (FPG) after previous CT exercise interventions of −4.0 mg/dL and a standard deviation of 1.0 mg/dL were obtained. Thus, a sample with a minimum of four cases per group (minimum sample of *n* = 8) gave us an alpha error of α = 0.05 and statistical power of β = 0.80. A total of *n* = 34 morbid obesity patients were recruited from the enrolment stage.

The eligibility criteria were as follows: (i) to be aged between 18 and 60 years, (ii) to be medically authorized to participate in the exercise programme, and (iii) to have a BMI ≥40 kg/m^2^, or BMI ≥35 kg/m^2^, with an associated comorbidities (i.e., diabetes, hypertension, or insulin resistance).The exclusion criteria were the following: (*i*) to have physical limitations that could restrict the performance of exercise (e.g., injuries to the musculoskeletal system), (*ii*) to have exercise-related dyspnoea or respiratory alterations, (*iii*) to have chronic heart disease with any worsening in the last month, and (*iv*) to show an adherence <80% to the total session interventions in the 12 months originally expected.

After enrolment (before the COVID-19 confinement), the participants were 1:1 randomly allocated to the following CT groups: HIIT + RT (started *n* = 17, dropouts *n* = 3 during training, analysed *n* = 14 until 20 weeks of intervention, and final analysed sample *n* = 7 until 20 months of physical inactivity due to the COVID-19 confinement), and to the RT + HIIT (started *n* = 17, dropouts *n* = 5 during training, analysed *n* = 12 until 20 weeks of intervention, and final analysed sample *n* = 11 until 20 months of physical inactivity due to COVID-19 confinement) (Figure 1). The clinical trial number registration is NCT04932642.

### 2.2. Metabolic Syndrome Outcomes

The MetS markers were screened using standard criteria [34]. All the participants were instructed to arrive at the laboratory following overnight fasting of 8 to 10 h, being measured between 08:00 and 9:00 in the morning. These conditions were taken at the baseline (Pre0) and post-intervention (Post1 and Post2). Blood samples of ~5 mL were taken to determine the MetS outcomes: FPG, high-density lipoprotein cholesterol (HDL-c), and triglycerides (Tg); additional markers, total cholesterol (Tc) and low-density lipoprotein cholesterol (LDL-c), were taken.

Systolic (SBP) and diastolic (DBP) blood pressure measurements were carried out according to the standard criteria [35]. Blood pressure was measured in the sitting position after 5 min of rest. Two recordings were made using an OMRON^TM^ digital electronic BP monitor (model HEM 7114, OMRON, Chicago, IL, USA), and the mean of these measurements was used for statistical analysis. Before taking these measurements, we informed the participants that they must not smoke or drink caffeine for at least 2 h prior to measurement. Additionally, we registered the heart rate at rest with the same equipment as secondary outcome.

The participants’ waist circumference (WC) was assessed with a tape in centimetres (Adult SECA^TM^, CA, USA) at the upper hipbone and the top of the right iliac crest, with a non-elastic measuring tape in a horizontal plane around the abdomen at the level of the iliac crest. The tape was snug but did not compress the skin and was parallel to the floor. The measurement was made at the end of a normal expiration [36].

### 2.3. Body Composition and Anthropometric Parameters

The body composition and anthropometric parameters were measured after fasting (>8 h). Body mass (kg), body fat (% and kg), skeletal muscle mass (kg), and lean mass (kg) were measured using a digital bio-impedance BIA scale (TANITA^TM^, model 331, Tokyo, Japan), and height (m) was measured using a SECA^TM^ stadiometer (model 214, Hamburg, Germany), with subjects in light clothing and without shoes. The BMI was calculated as the body mass divided by the square of the height (kg/m^2^). The BMI was determined to estimate the degree of obesity (kg/m^2^) using the standard criteria for the obesity and severe/morbid obesity classification [37]. Additionally, as the BIA equipment give us the information, we also reported the outcomes of bone mass, total body water and basal metabolic rate as secondary outcomes.

### 2.4. Six-Minute Walking Test

The day after the metabolic measurements, the physical condition of the participants in both groups was measured through endurance and muscle strength testing. First, a six-minute walking test (6 Mwt) was used to estimate cardiorespiratory fitness (CRF). The test was performed in an indoor court on a flat surface (30 m long), with two reflective cones placed at the ends to indicate the distance. During the test, an exercise physiologist assisted the participants with instructions [38].

### 2.5. Handgrip Strength

Handgrip strength (HGS) was assessed using a digital dynamometer (Baseline^TM^ Hydraulic Hand Dynamometers, NY, USA), which has been used in previous studies [39]. Two attempts were made, measuring each dominant and non-dominant arm, and the best result from each was selected and registered, as previously reported [39].

### 2.6. Concurrent Training Intervention

The CT programme had two sections of HIIT and RT, which were applied in different orders to the two experimental groups HIIT + RT and RT + HIIT. The two groups were proposed originally to test several unknown physiological adaptations from the ‘order’ configurations on MetS outcomes, until the COVID-19 pandemic started along with social confinement at home. Before the start of each exercise group, both HIIT + RT and RT + HIIT participants were involved in four familiarization sessions. In the HIIT + RT group, the HIIT section consisted of 60 s of maximum-intensity exercise using a magnetic resistance static bicycle (Oxford^TM^ Fitness, model BE-2701, Santiago, Chile), followed by 60–120 s of passive recovery over the bicycle, and it was repeated four to seven times according to the weekly schedule [40]. The intensity of the exercise was measured on the Borg scale of 1 to 10 for perceived exertion, and the participants worked at a level between 6 and 9 points.

Second, in the RT section, three out of four RT exercises were developed (according to the planning week), targeting the following different muscle groups: (1) forearm, (2) knee flexors and extensors, (3) trunk, (4) chest, (5) shoulder elevators, (6) horizontal shoulder flexors, (7) extensors, and, finally, (8) plantar flexors. The exercises alternated muscle groups for each session; for example, session 1 contained exercises for the 1, 3, and 5 muscle groups and session 2 involved exercises for the 4, 6, and 8 muscle groups. These exercises were performed in three sets of as many repetitions (continuous concentric/eccentric voluntary contraction) as possible in 60 s, followed by 60 to 120 s of passive recovery, as previously reported [41]. To estimate the intensity of work in the different RT exercises, the maximum dynamic muscular strength (1 RM) was estimated indirectly through the Brzycki formula [42], with fewer than 12 maximum repetitions. The RT + HIIT group performed the same training programme as the HIIT + RT group (described above) but the order of the HIIT and RT exercises was reversed (i.e., first RT and then HIIT).

### 2.7. Statistical Analyses

The data are presented as the mean and (±) standard deviation (SD). The normality and assumptions for all the data were checked using the Shapiro–Wilk test. Wilcoxon’s test was used for non-parametric data. The two-way ANOVA (groups × time) test was performed to test for differences between groups, comparing the baseline (Pre0), the final of the 20-week intervention in HIIT + RT, and T + HIIT (Post1) as well as after 20 months of COVID-19 confinement time (Post2). To identify the time difference, Sidak’s post hoc test was used. These analyses were carried out using the statistical Graph Pad Prism 8.0 software (Graph Pad Software, San Diego, CA, USA).

## 3. Results

### 3.1. Anthropometry and Body Composition (Secondary Outcomes)

There were no significant differences in anthropometric (body mass, BMI) and body composition (body fat in % and kg, lean mass, skeletal muscle mass, bone mass, total body water, and basal metabolic rate) outcomes between groups at the baseline (Table 1).

### 3.2. Training-Induced Effects on Anthropometrics, Body Composition, Cardiovascular, Metabolic, and Physical Fitness (Secondary Outcomes)

As training-induced effects, both the HIIT + RT and the RT + HIIT group did not elicit significant changes from Pre0 to Post1 in all the outcomes, with the exception of the RT + HIIT group for the outcome body mass (120.1 ± 21.8 vs. 115.1 ± 20.8 kg, *p* < 0.05). On the other hand, from the Post1 to the Post2 measurement, the HIIT + RT group showed significant changes in body fat as a percentage (48.0 ± 3.8 to 49.7 ± 3.2%, *p* < 0.05) and body fat in kg (49.9 ± 13.5 to 54.8 ± 12.9 kg, *p* < 0.05) (Table 1). In the same group, from Pre0 to Post2, there were significant changes in the outcomes of body mass (104.4 ± 20.3 to 109.2 ± 19.3 kg), body fat as a percentage (48.2 ± 4.2 to 49.7 ± 3.2%) and in kg (50.8 ± 13.6 to 54.3 ± 6.7 kg), lean mass (53.2 ± 7.1 to 54.3 ± 6.7 kg), SMM (50.5 ± 6.8 to 51.5 ± 6.4 kg), and basal metabolic rate (1688.0 ± 251.0 to 1727.0 ± 237.8 kcal), all *p* < 0.05 (Table 1). From the Post1 to the Post2 measurements, the RT + HIIT group showed significant changes in body mass (115.1 ± 20.8 to 121.8 ± 26.6 kg), lean mass (56.2 ± 4.2 to 62.0 ± 10.3 kg), SMM (53.3 ± 4.0 to 58.8 ± 9.8 kg), bone mass (2.8 ± 0.1 to 3.1 ± 0.4), total body water (40.2 ± 4.4 to 46.5 ± 9.4), and basal metabolic rate (1818.0 ± 175.8 to 1985.0 ± 349.0 kcal), all *p* < 0.05 (Table 1).

In the HIIT + RT group, there were significant differences from Post1 to Post2 in 6 Mwt (660.9 ± 104.3 to 504.5 ± 119.9 m, *p* < 0.05) (Table 2). From Pre0 to Post1, there were significant changes in 6 Mwt (540.9 ± 117.1 to 504.5 ± 119.9 m) (Table 2). In the RT + HIIT group, significant differences were found from Pre0 to Post1 in the heart rate at rest (78.6 ± 10.1 to 92.9 ± 17.8 beats/min), and, from Pre0 to Post2, there were significant changes in handgrip strength (29.5 ± 9.3 to 35.2 ± 9.2 kg) in participants’ non-dominant hand (Table 2).

### 3.3. Training-Induced Effects on Metabolic Syndrome Outcomes (Main Outcomes)

In the HIIT + RT group, WC showed significant changes from Pre1 to Post2 (115.0 to 118.1 cm, *p* = 0.035) (Figure 2, panel A), whereas, in the RT + HIIT group, WC showed significant changes from Pre0 to Post1 (126.4 to 121.6 cm, *p* = 0.028) (Figure 2, panel B). SBP showed significant changes from Post1 to Post2 (126.6 to 132.8 mmHg, *p* = 0.041) (Figure 2, panel C), whereas, in the RT + HIIT group, SBP showed significant changes from Pre0 to Post1 (142.2 to 135.0 mmHg, *p* = 0.026) (Figure 2, panel D). For DBP, the RT + HIIT group showed significant changes from Pre0 to Post1 (85.0 to 93.1 mmHg, *p* = 0.015) (Figure 2, panel F).

Tg in the HIIT + RT group showed significant changes from Pre0 to Post1 (131.1 to 91.0 mg/dL, *p* = 0.023) and from Post1 to Post2 (Figure 3, panel C), whereas Tg in the RT + HIIT group showed significant changes from Pre1 to Post2 (118.4 to 204.7 mg/dL, *p* < 0.0001) (Figure 3, panel F). There were no significant differences in HDL-c (Figure 3, panels A and B) and FGP in both groups (Figure 3, panels E and F) (*p* > 0.05).

## 4. Discussion

The aim of this study was to describe the effects of the imposed COVID-19 confinement on the cardiometabolic health benefits acquired through an exercise training programme that started before the COVID-19 pandemic in populations with morbid obesity but that was sharply interrupted at 20 weeks due to social confinement. The main results of this study are that (*i*) 20 months of COVID-19 social confinement worsened MetS outcomes, which improved with the 20-week training period of RT + HIIT, such as WC, SBP, and Tg in the RT + HIIT group, and (*ii*) there was a reduction in the endurance performance capacity in the HIIT + RT group after 20 months of obligated confinement.

Unfortunately, as HIIT + RT showed worse MetS outcomes during the 20 months of COVID-19 (i.e., WC, SBP (Figure 2), and Tg (Figure 3)), this does not mean that RT + HIIT has a better residual capacity to maintain or retain the beneficial exercise adaptations during the COVID-19 confinement due to this group’s increased (although non-significantly) WC, SBP, MAP, Tc, and LDL-c and decreased HDL-c and its performance in the 6 Mwt, the social confinement thus affecting both main MetS and secondary outcomes in this sample of participants with high cardiometabolic risk (Table 1).

Previous evidence has shown that pathologies derived from lockdown, isolations, and social distancing have similarly worsened cardiometabolic health [43,44]. This obligatory social confinement due to COVID-19, at present almost worldwide as a preventive measure, produces per se more sedentary behaviour, more opportunities for energy consumption, and other psychological impacts that promote stress, anxiety, and overeating, the subsequent modification of anthropometrics (i.e., increased body weight) and clinical health parameters (i.e., increased FPG, Tc, LDL-c, and others) being no more that the collateral/secondary effect [45,46]. The previously described increase in sedentary behaviour has also been associated with muscle mass loss and systemic inflammation, leading to higher cardiometabolic risk [47]. For example, in the present study (only considering significant training-induced changes from Pre0 to Post1), the RT + HIIT group members reduced WC (Δ − 4.8 cm) and SBP (Δ − 7.2 mmHg) but, after 20 months of confinement, they had increased results (Δ + 5.4 cm and Δ + 6.8 mmHg, respectively, for WC and SBP) (Figure 2). In other reports, after 12 weeks of supervised exercise training (i.e., endurance, RT, or CT), Timmons et al. [48] reported that, 12 months after exercise cessation, older adult participants (*n* = 53; 70.8 y) had an increased body fat percentage (Δ + 4.3%) and decreased lean mass (Δ − 0.6%), strength (leg press Δ − 5.6%; chest press Δ − 11.0%), and cognitive function (Δ − 3.7%), some of these results being different from those in the present study considering our maintenance of body fat percentage and lean mass, as can be seen in Table 1.

Interesting, Timmons et al. [48] did not find decreases in the handgrip muscle strength as well as in other specific outcomes related to functionality in older adults, such as gait velocity, sit to stand, and timed up-and-go tests. After 9 months of the exercise training programme in older adults (endurance exercise 15–25 min, RT 15–20 min), followed by 12 months of no exercise/detraining, Leitão et al. [49] reported that older adults with a significant loss in body weight of Δ − 1.9% during training recovered Δ + 0.64%, those with a decrease of Δ − 2.4% in body fat recovered Δ + 1.1%, those with a decrease of Δ − 5.1/− 5.2% in SBP/DBP recovered Δ + 7.8% in SBP/DBP, respectively, during the detraining period, those with a Δ − 16.4% decrease in Tg recovered Δ + 7.2%, and those with a Δ − 15.2% decrease in FPG after training had recovered Δ + 19.3% after the detraining period. Additionally, both upper- and lower-body muscle strength increased after the training period (Δ + 30.3/+ 30.6%); unfortunately, after 12 months of exercise cessation, both had worsened, participants losing Δ − 12.7/ − 11.6% of muscle strength in each upper- and lower-body compound, respectively.

Thus, the inactivity of the skeletal muscle was shown to be highly affected in some outcomes of the MetS, such as SBP/DBP, body fat percentage, and Tg, including muscle strength; however, some of these outcomes, in the present study, such as body fat percentage, could show more residual capacity (i.e., the capacity not to increase or change in line with being altered), together with the bone mass, total body water, and basal metabolic rate. In this contrast from our present study, we speculate that, due to those lean mass and skeletal muscle mass increases during the exercise period, the COVID-19 social confinement influenced all those participants who affectively reported better skeletal muscle progress to a lower degree, and thus this tissue could play a protective factor in the imposed condition of a sedentary state and physical inactivity.

The increased body weight and therefore BMI were a consequence of the dietary habits and sedentary behaviour during the lockdown period [50,51,52]. An increased fat mass percentage, WC, and abdominal perimeter were also observed [52,53,54]. According to the increases in the ratio of waist to hip circumferences, these effects were associated with increased central obesity [55]. Biochemical parameters were also affected: the lipid profile showed an increase in total cholesterol levels, which corresponded to an additional increase in LDL cholesterol levels and a decrease in HDL cholesterol levels, with statistically significant differences [45,56,57]. Plasma glucose levels deteriorated [56], probably in connection with the increased rate of obesity and being overweight and the decrease in physical exercise [58].

The present lockdown adversely affected multiple risk factors related to MetS. The plasma concentrations for LDL and HDL cholesterol increased and decreased, respectively. Concurrently, the blood glucose concentrations and blood pressure increased [56]. In patients with blood glucose levels in the range of diabetes mellitus, a statistically significant decrease in LDL cholesterol levels was detected, although this was not statistically significant in patients with prediabetes values, for whom there was no clear relationship with changes in LDL cholesterol values [56].

Regarding blood pressure levels, the lockdown also caused a deterioration in people who were not previously hypertensive, probably due to their lifestyle during these months and the worsening of the population’s health status owing to a change in dietary habits and physical activity [56]. Other studies have referred to these changes in blood pressure during the lockdown due to COVID-19 [45,59]. The social confinement during the COVID-19 pandemic was associated with an increase in severe arterial hypertension, independently of biological factors such as age or sex. In addition, people with hypertension have a more unfavourable evolutionary course of the disease when they contract COVID-19 [60,61]. Notably, since low HDL cholesterol, a larger WC, hyperglycaemia, hypertension, and hypertriglyceridemia are considered as global measures for cardiovascular disease risk and developing type 2 diabetes mellitus, it is relevant to increase the possibilities to extend the exercise training programmes from in-person to online platforms, of which technological/internet support for participants and exercise professionals can be key elements [62].

In general, until the training period in the present study, the HIIT + RT order of CT improved 1 out of 5 and worsened 3 out of 5 MetS outcomes after the COVID-19 confinement, whereas, in contrast, the RT + HIIT order of CT improved 2 out of 5 and worsened 1 out of 5 MetS outcomes after the 20 months of COVID-19 social confinement. This summary of the results from both orders of CT shows that it is difficult to examine the potential advantages or disadvantages of both training-induced capacity changes from one to the other order during the training period (i.e., from Pre0 to Post1 in 20 weeks (5 months)) as well as after the 20 months of the COVID-19 confinement due to this unexpected pandemic state, which sharply interrupted the exercise programme, increasing the possibilities from the environment to acquire unhealthy habits. In a recent report by Durão et al. [57], patients (*n* = 75) with morbid obesity who were exposed similarly to the COVID-19 confinement reported an increase in energy-dense, micronutrient-poor foods and thus a threefold increase in the odds ratio for an increased BMI (+0.8 kg/m^2^); additionally, the participants showed increased depression and anxiety symptoms. These results are in coherence with previous reports of changes in the physical activity practice in the Chilean population, in which ‘moderate-intensity’ physical activity decreased from 103.7 to 56.7 min/week, respectively, and ‘vigorous-intensity’ physical activity decreased from 49.9 to 26.7 min/week, respectively), this situation being, by contrast, reported with an increase in sitting time from 314 min/week before the COVID-19 confinement to 471.9 min/week and finally with increased screen time from 246.5 to 455.6 min/week [58]. In addition, despite the beneficial effects of both CT orders of HIIT + RT and RT + HIIT on morbidly obese patients, as well as previous literature that has reported the COVID-19 confinement in several populations [58,59]. In the present study, one of the major novel results is the confirmed hypothesis that 20 months of COVID-19 social confinement worsened several MetS outcomes, previously improved through a 20-week CT training period, in each CT order group and promoted detrimental effects on this cohort.

### Strengths and Limitations

The first strength of the present study is that we were conducting a normal exercise training programme with morbidly obese populations involved in two different CT orders when, at 20 weeks of intervention, the COVID-19 social confinement actions started, Post1 being rapidly applied. (*i*) We obtained information about a population with high cardiometabolic risk that started social confinement, and (*ii*) after another 20 months during the COVID-19 period, we obtained Post2 measurements, having the opportunity to report the detrimental effect of COVID-19 at the level of an exercise training programme applied to a particularly at-risk population. Furthermore, (*iii*) we included other additional anthropometric, body composition, and physical fitness tests to complement the CT order and COVID-19 effects. As the main limitation, unfortunately, the final sample size was reduced from the start of the social confinement for 20 months until the Post2 measurement.

## 5. Conclusions

The 20 months of social confinement due to COVID-19 worsened MetS outcomes that had improved from 20 weeks for RT + HIIT during the training period, such as WC, SBP, and Tg, and for HIIT + RT; worryingly, SBP increased to another more serious clinical diagnosis in both groups. These results revealed the practical need to promote and maintain physical activity or exercise training strategies in future at-home social confinement imposed due to pandemic viruses in populations with high cardiometabolic risk, such as patients with morbid obesity.

**New and Noteworthy:** The effects of concurrent training applied in different orders, such as high-intensity interval training plus resistance (HIIT + RT) or in reverse order (RT + HIIT), have provided relevant information for improving cardiometabolic health independent of their order of application in morbidly obese patients who are candidates for bariatric surgery. The uniqueness of the present study is the finding that COVID-19 social confinement worsened the previous concurrent training benefits acquired before the pandemic state in metabolic syndrome outcomes such as systolic blood pressure and plasma triglycerides. These findings claim for a need of a major capacity for translating exercise training programs from in person to an ‘online’ follow-up in order to maintain population health benefits if future social distance or confinement state.

## Figures and Tables

**Figure 1 ijerph-19-13408-f001:**
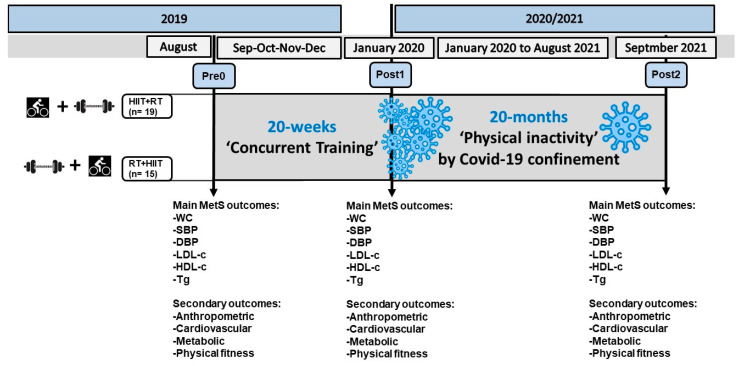
Study protocols.

**Figure 2 ijerph-19-13408-f002:**
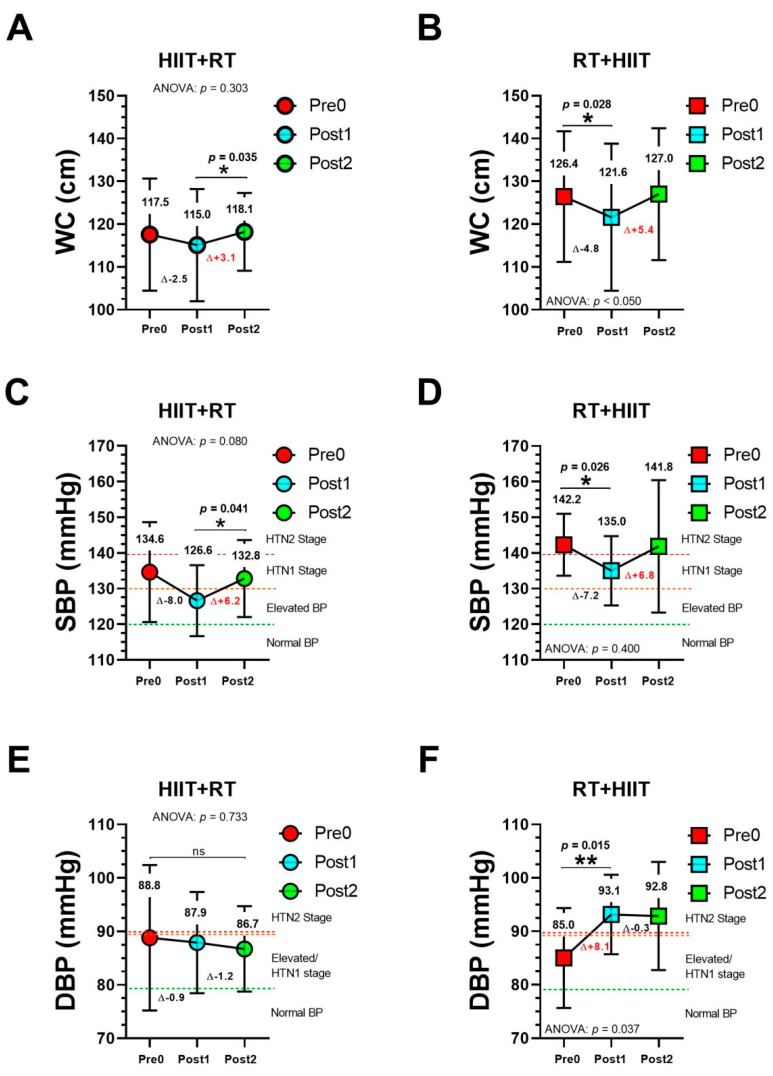
Metabolic syndrome outcomes (WC [panels **A**, **B**], SBP [panels **C**, **D**]/DBP [panels **E**, **F**]) after 20 weeks of different CT orders and after 20 months of COVID-19 confinement. (*) Denotes significant differences at level *p* < 0.05. (**) Denotes significant differences at level *p* < 0.001. (ns) Non-significant changes.

**Figure 3 ijerph-19-13408-f003:**
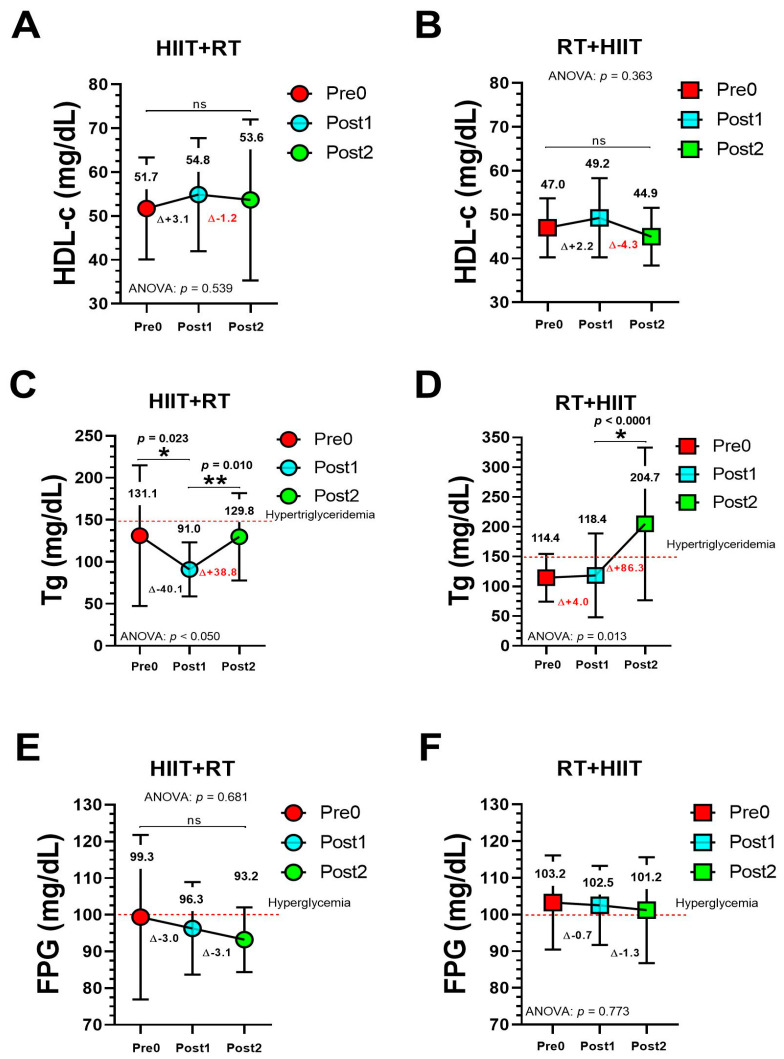
Metabolic syndrome outcomes (HDL-c [panels **A**, **B**], Tg [panels **C**, **D**], and FPG [panels **E**, **F**]) after 20 weeks of different CT orders and after 20 months of COVID-19 confinement. (*) Denotes significant differences at level *p* < 0.05. (**) Denotes significant differences at level *p* < 0.001. (ns) Non-significant changes.

**Table 1 ijerph-19-13408-t001:** Comparison between groups according to anthropometric and body composition variables.

	Time	HIIT + RT	RT + HIIT	Baseline *p*-Value
Age (y)		45.0 ± 8.9	35.5 ± 14.7	*p* = 0.108
**Anthropometric**				
Body mass (kg)	Pre0	104.4 ± 20.3	120.1 ± 21.8	*p* = 0.132
	Post1	102.7 ± 20.6	115.1 ± 20.8 ^¥^	
	Post2	109.2 ± 19.3 ^†^	121.8 ± 266 *^†^	
BMI (kg/m^2^)	Pre0	42.1 ± 6.6	47.5 ± 8.4	*p* = 0.153
	Post1	41.4 ± 6.4	45.6 ± 8.3	
	Post2	44.1 ± 6.5	47.6 ± 10.3 ^†^	
**Body composition**				
Body fat (%)	Pre0	48.2 ± 4.2	50.6 ± 3.7	*p* = 0.222
	Post1	48.0 ± 3.8	50.7 ± 4.8	
	Post2	49.7 ± 3.2 *^†^	48.4 ± 4.3	
Body fat (kg)	Pre0	50.8 ± 13.6	61.4 ± 15.9	*p* = 0.164
	Post1	49.9 ± 13.5	59.3 ± 16.3	
	Post2	54.8 ± 12.9 *^†^	59.7 ± 18	
Lean mass (kg)	Pre0	53.2 ± 7.1	58.5 ± 6.6	*p* = 0.0.47
	Post1	52.6 ± 7.3	56.2 ± 4.2	
	Post2	54.3 ± 6.7 ^†^	62.0 ± 10.3 *^†^	
Skeletal muscle mass (kg)	Pre0	50.5 ± 6.8	55.6 ± 6.2	*p* = 0.129
	Post1	50.0 ± 6.9	53.3 ± 4.0	
	Post2	51.5 ± 6.4 ^†^	58.8 ± 9.8 *^†^	
Bone mass (kg)	Pre0	2.7 ± 0.3	2.9 ± 0.3	*p* = 0.157
	Post1	2.6 ± 0.3	2.8 ± 0.1	
	Post2	2.7 ± 0.3	3.1 ± 0.4 *^†^	
Total body water (%)	Pre0	38.7 ± 5.9	41.8 ± 4.5	*p* = 0.259
	Post1	37.3 ± 5.4	40.2 ± 4.4	
	Post2	39.7 ± 5.6	46.5 ± 9.4 *^†^	
Basal metabolic rate (kcal)	Pre0	1688.0 ± 251.0	1897.0 ± 230.0	*p* = 0.094
	Post1	1670.0 ± 255.1	1818.0 ± 175.8	
	Post2	1727.0 ± 237.8 ^†^	1985.0 ± 349.0 *^†^	

Data are shown in mean and ± standard deviation. Times are described as follows: Pre0 is the baseline measurements; Post1 is the post-20 weeks of concurrent training intervention; Post2 is the post-20 months of physical inactivity due to COVID-19 confinement measures. **Groups are described as follows**: HIIT + RT is the high-intensity interval training plus resistance training group; RT + HIIT is the resistance training plus high-intensity interval training group. **Outcomes are described as follows**: BMI is the body mass index. Within-group analyses were tested using two-way ANOVA (groups–time). ^¥^ denotes significant differences between Pre0 and Post1 at *p* < 0.05 according to the Sidak post hoc test. * denotes significant differences between Pre0 and Post2 at *p* < 0.05 according to the Sidak post hoc test. ^†^ denotes significant differences between Post1 and Post2 at *p* < 0.05 according to the Sidak post hoc test.

**Table 2 ijerph-19-13408-t002:** Comparison between groups according to anthropometric cardiovascular, metabolic, and fitness parameters.

	Time	HIIT + RT	RT + HIIT	*p-*Value
**Cardiovascular**				
Heart rate resting (beats/min)	Pre0	85.2 ± 17.8	78.6 ± 10.1	*p* = 0.257
	Post1	86.9 ± 16.9	92.9 ± 17.8 ^¥^	
	Post2	85.2 ± 10.8	84.2 ± 16.1	
**Metabolic**				
Total cholesterol (mg/dL)	Pre0	177.6 ± 33.7	191.6 ± 23.5	*p* = 0.356
	Post1	180.6 ± 34.2	170.5 ± 26.9	
	Post2	180.5 ± 38.6	178.3 ± 19.6	
LDL cholesterol (mg/dL)	Pre0	116.8 ± 39.1	124.6 ± 17.2	*p* = 0.630
	Post1	116.6 ± 20.9	112.7 ± 22.8	
	Post2	123.5 ± 35.8	125.0 ± 20.1	
**Physical fitness**				
6 Mwt (m)	Pre0	540.9 ± 117.1	531.4 ± 51.7	*p* = 0.843
	Post1	660.9 ± 104.3	585.0 ± 65.9	
	Post2	504.5 ± 119.9 *^†^	541.4 ± 94.0	
Handgrip strength dominant (kg)	Pre0	29.1 ± 6.5	32.2 ± 9.6	*p* = 0.425
	Post1	31.6 ± 6.4	33.5 ± 4.9	
	Post2	28.5 ± 7.1	33.8 ± 7.3	
Handgrip strength non-dominant (kg)	Pre0	26.5 ± 6.3	29.5 ± 9.3	*p* = 0.425
	Post1	29.8 ± 6.7	31.2 ± 5.9	
	Post2	27.7 ± 7.2	35.2 ± 9.2 *	

Data are shown in mean and ± standard deviation. **Times are described as follows**: Pre0 is the baseline measurements with HIIT + RT *n* = 17 and RT + HIIT *n* = 17; Post1 is the post-20 weeks of concurrent training intervention measurements with HIIT + RT *n* = 14 and RT + HIIT *n* = 12; Post2 is the post-20 months of physical inactivity due to COVID-19 confinement measures with HIIT + RT *n* = 11 and RT + HIIT *n* = 7. **Groups are described as follows**: HIIT + RT is the high-intensity interval training plus resistance training group; RT + HIIT is the resistance training plus high-intensity interval training group. **Outcomes are described as follows**: LDL-c is low-density lipids; 6 Mwt is the six-minute walking test. ^†^ indicates an analysis using one-way ANOVA at *p* < 0.05. Within-group analyses were tested by two-way ANOVA (groups–time). ^¥^ denotes significant differences between Pre0 and Post1 at *p* < 0.05 according to the Sidak post hoc test. * denotes significant differences between Pre0 and Post2 at *p* < 0.05 according to the Sidak post hoc test.

## Data Availability

Not applicable.

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
