# Peer review of "The Social Distance Impacts from COVID-19 Pandemic on the Development of Two Orders of a Concurrent Training Programme for Morbidly Obese Patients"

_ijerph, 2022, doi:10.3390/ijerph192013408_

Round 1

Reviewer 1 Report

Some suggestions after I reviewed your article:

1. The abstract must be unstructured and up to 200 words, according to Instructions for authors.

2. In material and Methods at line 130 you mentioned BMI less than 40 (at eligibility criteria), but in Table 1 BMI of the subjects is between 41.4 and 47.6. Where is the error?

3. At Table 2 for arterial pressure I recommend to present systolic and diastolic values not the mean of both.

4. In Figure 2.E - check if the values are correct, especially for post2.

Author Response

Reviewer 1

  1. The abstract must be unstructured and up to 200 words, according to Instructions for authors.

Response; dear reviewer you are right, however according another reviewer have given us more information, so we have more tnah 200 words in abstract section.

  1. In material and Methods at line 130 you mentioned BMI less than 40 (at eligibility criteria), but in Table 1 BMI of the subjects is between 41.4 and 47.6. Where is the error?

Response; Dear reviewer we committed a mistake, we have changed it;

The eligibility criteria were as follows: (i) to be aged between 18 and 60 years, (ii) to be medically authorized to participate in the exercise programme, and (iii) to have a BMI≥40 kg/m2, or BMI≥35 kg/m2, with an associated comorbidities (i.e., diabetes, hypertension, or insulin resistance).

  1. At Table 2 for arterial pressure I recommend to present systolic and diastolic values not the mean of both.

Response; done  

  1. In Figure 2.E - check if the values are correct, especially for post2.

Response: Thank you for your comment, the values are corrected.

Reviewer 2 Report

INTRODUCTION

- The introduction is well written and provides a suitable background and covers the research topic of the manuscript.

- However, I suggest reporting that up to line 74 reference is being made to the effects of COVID-19 of different populations around the world. Please consider to cite the following papers: 1) "Gjaka M, et al. (2021). The effect of COVID-19 lockdown measures on physical activity levels and sedentary behaviour in a relatively young population living in Kosovo. Journal of clinical medicine, 10(4), 763." ; 2) "Mauro M, et al. (2022). Effects of quarantine on Physical Activity prevalence in Italian Adults: a pilot study. PeerJ 10:e14123". So to be introduced immediately after the Latin American population (line 74).

- Please define "Lifestyle exercise training programmes".

- (line 89-92): "Along this line,…and actions abruptly started" Authors should provide some references for this statement. For example: 1) "Polero P, et al. Physical Activity Recommendations during COVID-19: Narrative Review. Int J Environ Res Public Health. 2020;18(1):65" ; 2) "Bentlage E, et al. Practical Recommendations for Maintaining Active Lifestyle during the COVID-19 Pandemic: A Systematic Literature Review. Int J Environ Res Public Health. 2020;17(17):6265".

MATERIALS AND METHODS

- Please rename paragraph 2.1, authors should replace "subjects" with "participants".

- Authors should specify the process and type of randomization used to allocate participants in the two groups.

- As reported by the authors (line 207-209), the two programs are the same and only the order changes (i.e., HIIT+RT vs RT+HIIT). Please specify it clearly also in the abstract and at the beginning of the methods section during the description of the study design.

- It is unclear how the recruitment took place before COVID-19. Where did the participants come from? What was the sampling method? Please describe in detail the process.

- (line 114-118): The authors reported that the number of participants "reduced" for both groups. Do they mean they have had dropouts?

- (line 137-141): This information has already been reported (line 114-118). Please revise this section.

- Methods are quite confused. Eligibility criteria should be reported prior to the study setting and intervention description. Similarly, the intervention should be reported before the outcomes. Please follow the consort checklist for randomised trials. Here the link: https://www.consort-statement.org/

RESULTS

- In the "statistical analysis" section authors reported that they used both parametric and non-paramteric test. However, it is not clear which parameters were normally distributed. Authors should report, among the results which test was used for each parameter. Or they should specify which were the non-parametric data.

- Although among the results of "body composition and anthropometric parameters" the authors describe the total body water and the basal metabolic rate parameter, they did not report in the methods section (par 2.3) of having recorded these data. Please review.

- Similarly, it does not seem to me that the authors have reported that they have recorded the heart rate at rest, but it is described in the results. Please double check the entire manuscript.

DISCUSSION

- In the discussion the authors should report whether their hypothesis has been confirmed or not.

- What are the practical implications of the study?

Author Response

INTRODUCTION

- The introduction is well written and provides a suitable background and covers the research topic of the manuscript.  However, I suggest reporting that up to line 74 reference is being made to the effects of COVID-19 of different populations around the world. Please consider to cite the following papers: 1) "Gjaka M, et al. (2021). The effect of COVID-19 lockdown measures on physical activity levels and sedentary behaviour in a relatively young population living in Kosovo. Journal of clinical medicine, 10(4), 763." ; 2) "Mauro M, et al. (2022). Effects of quarantine on Physical Activity prevalence in Italian Adults: a pilot study. PeerJ 10:e14123". So to be introduced immediately after the Latin American population (line 74).

Response: we have adapted and added the references

-In addition, around the word, it has been reported decreases in PA patterns [24, 25].  Likewise, another study reported that reductions of PA was more pronounced in Latin America compared with European countries [10]. Considering that Latin America is a region characterized by wide social inequalities, poor healthcare access, and a high prevalence of cardiometabolic diseases, such as type 2 diabetes mellitus and arterial hypertension [26], reporting information from those populations that are more affected by COVID-19 and face higher MetS risk could increase the possibilities of implementing preventive strategies.

Response:  Dear reviewer, thanks for this comment. According to this sentence, we have now re-written and improved the clarity of this paragraph as follows;

“…In addition, around the world, it has been reported a sharp decrease in the PA patterns globally [24, 25]. However, worryingly early evidence from the start of the pandemic stated reported that these reductions in PA were more pronounced in Latin America compared with European countries [10]. Thus, considering that Latin America is a region characterized by wide social inequalities, poor healthcare access, and a high prevalence of cardiometabolic diseases, such as type 2 diabetes mellitus and arterial hypertension [26], reporting information from those populations that are more affected by COVID-19 and face higher MetS risk could increase the possibilities of implementing preventive strategies in a future post-pandemic state..….”

-Lifestyle exercise training programmes are characterized for the implementation of regular exercise interventions to modify specific unhealthy lifestyle components ( in-sufficient PA) [27, 28].

- Please define "Lifestyle exercise training programmes".

Response: Dear reviewer, thanks by the comment. Attending to this, we have now clarified the sentence as follows:

“…A lifestyle change program is characterized by the inclusion of regular exercise, and additionally other specific behaviours such as nutrition, and sleep patterns, among others (tobacco, and alcohol consumption) [27, 28]. Regular exercise plays …”

- (line 89-92): "Along this line,…and actions abruptly started" Authors should provide some references for this statement. For example: 1) "Polero P, et al. Physical Activity Recommendations during COVID-19: Narrative Review. Int J Environ Res Public Health. 2020;18(1):65" ; 2) "Bentlage E, et al. Practical Recommendations for Maintaining Active Lifestyle during the COVID-19 Pandemic: A Systematic Literature Review. Int J Environ Res Public Health. 2020;17(17):6265".

Response: Dear reviewer, thanks by the references. Following your recommendation, we have now incorporated both at the final of the commented paragraph.

 MATERIALS AND METHODS

- Please rename paragraph 2.1, authors should replace "subjects" with "participants".

Response: Dear reviewer, it was changes appropriately.

- Authors should specify the process and type of randomization used to allocate participants in the two groups.

Response: Dear reviewer, thanks by the comment. Now we have incorporated this information as follows in the “Methods” section;

“…After enrolment (before the COVID-19 confinement), the participants were 1:1 randomly allocated to the following CT groups: HIIT+RT (..”

- As reported by the authors (line 207-209), the two programs are the same and only the order changes (i.e., HIIT+RT vs RT+HIIT). Please specify it clearly also in the abstract and at the beginning of the methods section during the description of the study design.

Response: Thanks by the comment. We have now clarified this point in both sections as follows;

Abstract:

“…Methods: This was an experimental randomized clinical study, in which sedentary morbidly obese women were assigned 1:1 to a high-intensity interval training (HIIT) plus resistance training (RT) group (HIIT+RT; n=11; BMI 42.1±6.6) or to the same exercise dose, but in different order group of RT plus HIIT group (RT+HIIT; n=7; BMI 47.5±8.4). Both groups undertook two sessions/week. W…”

Methods:

“…participate. The participants were allocated to the same exercise dose of CT high-intensity interval training plus resistance training (i.e., HIIT+RT) or to another group but of different order of resistance training plus high-intensity interval training (RT+HIIT). Both …”

- It is unclear how the recruitment took place before COVID-19. Where did the participants come from? What was the sampling method? Please describe in detail the process.

Response: Dear reviewer, thanks by the comment. We are plenty agree with you, and we have now re-ordered this section of the “methods” as follows;

“…2. Materials and Methods

2.1. Participants

This study was a randomized controlled trial in which (n=34) women with morbid obesity from the Morbid Obesity Association of Temuco City, Chile, were invited to participate by a public call and contacted by phone directly to all those interested. The participants were invited to participate in the interventions since they were part of a previous recommended lifestyle exercise programme in the same institution [30]. All the participants were informed of the pre–post procedures and of the possible risks/benefits potentially involved in participation in the study, after which they signed an informed consent form. The study was undertaken in accordance with the Declaration of Helsinki (2013) and has been approved by the Ethical Committee of the Universidad de La Frontera, Temuco, Chile (ACTA Nº 080_21).

The sample size was calculated using the G*Power software, and the observed delta changes in fasting plasma glucose (FPG) after previous CT exercise interventions of −4.0 mg/dL and a standard deviation of 1.0 mg/dL were obtained. Thus, a sample with a minimum of four cases per group (minimum sample of n=8) gave us an alpha error of α=0.05 and statistical power of β=0.80. A total of n=34 morbid obesity patients were recruited from the enrolment stage.

The eligibility criteria were as follows: (i) to be a candidate for bariatric surgery, (ii) to be aged between 18 and 60 years, (iii) to be medically authorized to participate in the exercise programme, and (iv) to have a mean BMI≥40 kg/m2, with/without associated comorbidities (i.e., diabetes, hypertension, or insulin resistance). The exclusion criteria were the following: (i) to have physical limitations that could restrict the performance of exercise (e.g., injuries to the musculoskeletal system), (ii) to have exercise-related dyspnoea or respiratory alterations, (iii) to have chronic heart disease with any worsening in the last month, and (iv) to show an adherence <80% to the total session interventions in the 12 months originally expected.

After enrolment (before the COVID-19 confinement), the participants were 1:1 randomly allocated to the following CT groups: HIIT+RT (started n=17, dropouts n=3 during training, analysed n=14 until 20 weeks of intervention, and final analysed sample n=7 until 20 months of physical inactivity due to the COVID-19 confinement), and to the RT+HIIT (started n=17, dropouts n=5 during training, analysed n=12 until 20 weeks of intervention, and final analysed sample n=11 until 20 months of physical inactivity due to COVID-19 confinement) (Figure 1). The clinical trial number registration is NCT04932642.….”

- (line 114-118): The authors reported that the number of participants "reduced" for both groups. Do they mean they have had dropouts?

Response: Dear reviewer, thanks by this comment. According with this we have now clarified this paragraph as follows;

“…After enrolment (before the COVID-19 confinement), the participants were 1:1 randomly allocated to the following CT groups: HIIT+RT (started n=17, dropouts n=3, analysed n=14 until 20 weeks of intervention, and final analysed sample n=7 until 20 months of physical inactivity due to the COVID-19 confinement). To the RT+HIIT (started n=17, dropouts n=5, analysed n=12 until 20 weeks of intervention, and final analysed sample n=11 until 20 months of physical inactivity due to COVID-19 confinement). …”

- (line 137-141): This information has already been reported (line 114-118). Please revise this section.

Response: Thanks dear reviewer, we have now removed this repeated information.

- Methods are quite confused. Eligibility criteria should be reported prior to the study setting and intervention description. Similarly, the intervention should be reported before the outcomes. Please follow the consort checklist for randomised trials. Here the link: https://www.consort-statement.org/

Response: Thanks by the comment. We think that with the previous asked comment where we re-ordered this section, this information has been clarified.

RESULTS

- In the "statistical analysis" section authors reported that they used both parametric and non-paramteric test. However, it is not clear which parameters were normally distributed. Authors should report, among the results which test was used for each parameter. Or they should specify which were the non-parametric data.

Response:

- Although among the results of "body composition and anthropometric parameters" the authors describe the total body water and the basal metabolic rate parameter, they did not report in the methods section (par 2.3) of having recorded these data. Please review.

Response: Dear reviewer, thanks by see this point. Following this comment, we have now added this information as follows;

“…2.3. Body Composition and Anthropometric Parameters

The body composition and anthropometric parameters were measured after fasting (>8 h). Body mass (kg), body fat (% and kg), skeletal muscle mass (kg), and lean mass (kg) were measured using a digital bio-impedance BIA scale (TANITATM, model 331, Tokyo, Japan), and height (m) was measured using a SECATM stadiometer (model 214, Hamburg, Germany), with subjects in light clothing and without shoes. The BMI was calculated as the body mass divided by the square of the height (kg/m2). The BMI was determined to estimate the degree of obesity (kg/m2) using the standard criteria for the obesity and severe/morbid obesity classification [37]. Additionally, as the BIA equipment give us the information, we also reported the outcomes of bone mass, total body water and basal metabolic rate as secondary outcomes...”

- Similarly, it does not seem to me that the authors have reported that they have recorded the heart rate at rest, but it is described in the results. Please double check the entire manuscript.

Response: Dear reviewer, thanks by the comment. Following this, we have now incorporated this information into the “..2.2. Metabolic Syndrome Outcomes..”, as follows;

“…Systolic (SBP) and diastolic (DBP) blood pressure measurements were carried out according to the standard criteria [35]. Blood pressure was measured in the sitting position after 5 minutes of rest. Two recordings were made using an OMRONTM digital electronic BP monitor (model HEM 7114, Chicago, IL, USA), and the mean of these measurements was used for statistical analysis. Before taking these measurements, we informed the participants that they must not smoke or drink caffeine for at least 2 hours prior to measurement. Additionally, we registered the heart rate at rest with the same equipment as secondary outcome.…”

DISCUSSION

- In the discussion the authors should report whether their hypothesis has been confirmed or not.

Response: Thanks dear reviewer. In line 396 to 400 was now reported this information, as follows;

“…In the present study, one of the major novel results is the confirmed hypothesis that 20 months of COVID-19 social confinement worsened several MetS outcomes, previously improved through a 20-week of CT training period, in each CT order group and promoted detrimental effects on this cohort.  …”

- What are the practical implications of the study?

Response: Thanks dear reviewer. After checking the “Instructions For Authors” of the journal, no practical implications section were found by our team. However, into the conclusion section, we highlight the practical implications following our findings, as follows;

“…The 20 months of social confinement due to COVID-19 worsened MetS outcomes that had improved from 20 weeks for RT+HIIT during the training period, such as WC, SBP, and Tg, and for HIIT+RT; worryingly, SBP increased to another more serious clinical diagnosis in both groups. These results revealed the practical need to promote and maintain physical activity or exercise training strategies in future at-home social confinement imposed due to pandemic viruses in populations with high cardiometabolic risk, such as patients with morbid obesity.…”

Round 2

Reviewer 2 Report

Congratulations to the authors for the revisions made. I have no other comments from my side.